# Enhanced Intestinal Absorption and Pharmacokinetic Modulation of Berberine and Its Metabolites through the Inhibition of P-Glycoprotein and Intestinal Metabolism in Rats Using a Berberine Mixed Micelle Formulation

**DOI:** 10.3390/pharmaceutics12090882

**Published:** 2020-09-17

**Authors:** Mihwa Kwon, Dong Yu Lim, Chul Haeng Lee, Ji-Hyeon Jeon, Min-Koo Choi, Im-Sook Song

**Affiliations:** 1College of Pharmacy and Research Institute of Pharmaceutical Sciences, Kyungpook National University, Daegu 41566, Korea; mihwa_k@naver.com (M.K.); kei7016@naver.com (J.-H.J.); 2College of Pharmacy, Dankook University, Cheon-an 31116, Korea; twins3639@naver.com (D.Y.L.); hang1130@naver.com (C.H.L.)

**Keywords:** berberine, pluronic P85 (P85), tween 80, pharmacokinetics, P-gp inhibition, CYP inhibition

## Abstract

We aimed to develop a berberine formulation to enhance the intestinal absorption and plasma concentrations of berberine through the inhibition of P-glycoprotein (P-gp)-mediated efflux and the intestinal metabolism of berberine in rats. We used pluronic P85 (P85) and tween 80, which have the potential to inhibit P-gp and cytochrome P450s (i.e., CYP1A2, 2C9, 2C19, 2D6, and 3A4). A berberine-loaded mixed micelle formulation with ratios of berberine: P85: tween 80 of 1:5:0.5 (*w/w/w*) was developed. This berberine mixed micelle formulation had a mean size of 12 nm and increased the cellular accumulation of digoxin via P-gp inhibition. It also inhibited berberine metabolism in rat intestinal microsomes, without significant cytotoxicity, up to a berberine concentration of 100 μM. Next, we compared the pharmacokinetics of berberine and its major metabolites in rat plasma following the oral administration of the berberine formulation (50 mg/kg) in rats with the oral administration of berberine alone (50 mg/kg). The plasma exposure of berberine was significantly greater in rats administered the berberine formulation compared to rats administered only berberine, which could be attributed to the increased berberine absorption by inhibiting the P-gp-mediated berberine efflux and intestinal berberine metabolism by berberine formulation. In conclusion, we successfully prepared berberine mixed micelle formulation using P85 and tween 80 that has inhibitory potential for P-gp and CYPs (CYP2C19, 2D6, and 3A4) and increased the berberine plasma exposure. Therefore, a mixed micelle formulation strategy with P85 and tween 80 for drugs with high intestinal first-pass effects could be applied to increase the oral absorption and plasma concentrations of the drugs.

## 1. Introduction

Berberine (Figure 1), an isoquinoline alkaloid isolated from Coptidis Rhizoma, has long been used as a folklore remedy in China for the treatment of bacteria-associated diarrhea and other gastrointestinal infections [1]. Berberine has several potential beneficial effects on many diseases associated with blood glucose and lipid metabolism. In one study, berberine lowered total cholesterol by 29% and decreased low-density lipoprotein (LDL) cholesterol by 25% and triglycerides by 35% in 32 hypercholesterolemic patients [2]. Berberine also has hypoglycemic effects in addition to its hypolipidemic effects [3]. Yin et al. reported that berberine showed a similar efficacy to lower blood glucose at an equivalent dose of metformin, glipizide, or rosiglitazone and outperformed it in lowering cholesterol and triglycerides [4,5]. Dong et al. performed a systematic review and meta-analysis of 14 randomized clinical studies on 1068 participants [6]. The results showed that berberine was effective as an antidiabetic therapeutic but was not superior to commercial oral hypoglycemics. Combination therapy of berberine and other oral hypoglycemics has been shown to significantly improve clinical outcomes, such as the fasting plasma glucose, glycosylated hemoglobin levels, fasting insulin levels, and triglyceride levels [6]. However, the low bioavailability (0.68% in rats) of berberine is one of the major obstacles for its therapeutic use [7]. Extensive intestinal first-pass effects are possibly responsible for this low plasma level. Liu et al. reported that 99.5% of berberine in oral doses disappeared during the gastrointestinal first-pass elimination process [8]. The area under concentration curve (AUC) of the berberine post intraportal vein administration accounts for 72% of its intravenous AUC. However, berberine’s AUC value post intraduodenal administration was nearly 0.5% of its AUC value post intraportal vein administration. These results suggest that first-pass elimination occurs predominantly in the rat intestine [8]. Recently, a total of 16 metabolites of berberine, 10 phase I metabolites, and 6 phase II metabolites were identified in the plasma, feces, bile, and other tissue samples of rats [9]. Among these metabolites, berberrubine, thalifendine, demethyleneberberine, and jatrorrhizine had relatively high plasma levels [10,11] and showed biological activities for antioxidative, hepatoprotective, and hypolipidemic effects [9]. Since berberine was metabolized in the intestine as well as in the liver, the formation of major berberine metabolites varied depending on its dosing route. That is, berberine and its major metabolites were all detected in plasma samples from rats after the intravenous injection of berberine (4 mg/kg), however berberine, berberrubine, and thalifendine were detected in plasma samples from rats after the oral administration of berberine at 100 mg/kg [8]. These results suggest that the involvement of the intestinal first-pass metabolism of berberine should be differentiated from the role of hepatic metabolism and the resultant metabolites. P-glycoprotein (P-gp) was also involved in the low bioavailability of berberine. The basal to apical transport of berberine was much greater than theapical to basal transport of berberine, and was significantly decreased by the treatment of cyclosporine A, a representative inhibitor of P-gp [12,13]. The treatment of known P-gp inhibitors, such as cyclosporine A, verapamil, and C219 antibodies, increased the berberine uptake in Caco-2 cells [13,14]. This suggests that P-gp inhibitor treatments might affect the berberine absorption process and pharmacokinetic profiles of berberine. The use of pharmaceutical excipients that modulate P-gp-mediated efflux or intestinal metabolism could be used to increase the bioavailability and plasma exposure of berberine.

In recent years, various pharmaceutical excipients have emerged, not only as solubilizing agents, but also as potential alternatives to P-gp and metabolic inhibitors [15,16]. For instance, pluronic triblock copolymers containing hydrophilic poly(ethylene oxide) (PEO) and hydrophobic poly(propylene oxide) (PPO) blocks (PEO–PPO–PEO) have been reported to reverse P-gp-mediated efflux in multidrug-resistant cancer cell lines [17,18]. Polyethylene glycol 400 (PEG400), pluronic P85 (P85), and vitamin E-D-α-tocopheryl polyethylene glycol 1000 succinate (TPGS) have been reported to inhibit in vitro P-gp-mediated efflux and intestinal metabolism when assessed using digoxin and verapamil as a substrate for P-gp and cytochrome P450 (CYP) 3A [19]. Similar results for the inhibitory effect of P85 on the P-gp-mediated efflux and for the increased bioavailability of P-gp substrate drugs have been reported [17,18,20,21,22]. Additionally, the use of tween 80 increased the oral bioavailability of digoxin through the inhibition of the P-gp-mediated intestinal efflux of digoxin [23] and inhibited the rat intestinal CYP3A activity [24]. The oral bioavailability and intestinal permeability of resveratrol were enhanced by nanoemulsion using labrasol and pluronic F68, which mainly inhibited intestinal glucuronidation [25]. There have been many reports on berberine formulation for the enhancement of oral bioavailability. Gui et al. [10] reported that the bioavailability of the oral berberine-loaded microemulsion formulation using tween 80 and PEG400 was 6.47-fold greater than that of the berberine tablet suspensions in rats. Chen et al. [7] used TPGS as an absorption enhancer to improve the C_max_ and AUC of berberine by 2.9- and 1.9-fold, respectively. A self-microemulsifying drug delivery system using labrasol increased the C_max_ and AUC of berberine in rats by 1.63- and 1.54-fold, respectively, compared with those of commercial tablets [25], although the underlying mechanisms for the increased C_max_ and AUC of berberine in the previous reports were not investigated. Based on these reports, the purpose of this study was to formulate berberine with the pharmaceutical excipients that modulate P-gp function and intestinal metabolism, and to investigate the bioavailability and pharmacokinetic alterations of berberine and its major metabolites in rats, in order to understand the role that pharmaceutical excipients play in bioavailability enhancement and pharmacokinetic modulation.

## 2. Materials and Methods

### 2.1. Materials

Berberine chloride, pluronic P85 (P85), tween 80, Hanks’ Balanced Salts Solution (HBSS), cyclosporine A, β-Glucuronidase from Helix pomatia (containing 300KU β-Glucuronidase and 10KU sulfatase/g), sodium acetate, acetaminophen, quinidine, ketoconazole, propranolol, diclofenac, hydroxydiclofenac, and KH_2_PO_4_ were purchased from Sigma-Aldrich chemicals (St. Louis, MO, USA). Phenacetin, S-mephenytoin, hydroxymephenytoin, dextromethorphan, dextrorphan, α-Naphthoflavone, sulfaphenazole, and S-benzylnirvanol were purchased from Toronto Research Chemicals (Toronto, ON, Canada). Midazolam and hydroxymidazolam were purchased from Cayman Chemical (Ann Arbor, MI, USA). Berberrubine (purity >98%) was purchased from ChemFaces (Wuhan, China). Thalifendine (purity >98%) was purchased from Shanghai Hekang Biotechnology Co., Ltd. (Shanghai, China). [^3^H]Digoxin (1.103 TBq/mmol) was purchased from Perkin Elmer Inc. (Boston, MA, USA).

Caco-2 cells (passage number 38–42) were purchased from American Type Culture Collection (Rockville, MD, USA). LLC-PK1-P-gp cells (passage number 22–23), ultrapooled human liver microsomes, reduced nicotinamide adenine dinucleotide phosphate (NADPH) generating solution, and fetal bovine serum were purchased from BD-Corning (Corning, NY, USA). Rat intestinal microsomes from SD rats were purchased from XenoTech (Kansas City, KS, USA). All other reagents were of reagent grade.

### 2.2. Effect of P85 and Tween 80 on the 5 CYP Activities

To examine the effects of P85 and tween 80 on the 5 CYP activities, the metabolites of CYP substrates were measured in the presence of P85 (0.1 and 1%) and tween 80 (0.1 and 1%). Briefly, the final concentrations for the 5 CYPs substrates were as follows: 50 μM of phenacetin (CYP1A2), 10 μM of diclofenac (CYP2C9), 100 μM of mephenytoin (CYP2C19), 5 μM of dextromethorphan (CYP2D6), and 5 μM of midazolam (CYP3A). This was prepared by adding 8.5 μL of 10-fold-concentrated methanol solution into reaction tubes, the methanol was dried using a vacuum concentrator, and 65 μL of 0.1 M KH_2_PO_4_ (pH 7.4) buffer was added to dissolve the substrate. Aliquots (20 μL) of 0.1 M KH_2_PO_4_ (pH 7.4) buffer containing 0.25 mg/mL of liver microsomal protein and aliquots (5 μL) of 0.1 M KH_2_PO_4_ (pH 7.4) buffer containing P85 (2 and 20%), tween 80 (2 and 20%), or representative inhibitors of 5 CYPs (20-fold working solution dissolved in 2% DMSO) were added to the substrate solution and preincubated for 5 min. The reaction mixtures were further incubated for 15 min at 37 °C in a thermoshaker, after a NADPH generating system (10 μL) had been added. The final concentration of the 5 CYPs inhibitors were as follows: 0.1 μM of α-Naphthoflavone (CYP1A2), 5 μM of sulfaphenazole (CYP2C9), 5 μM of S-benzylnirvanol (CYP2C19), 5 μM of quinidine (CYP2D6), and 5 μM of ketoconazole (CYP3A), and the final DMSO content was 0.1%. 

All the reactions were terminated by the addition of 50 μL of cold acetonitrile containing propranolol (10 ng/mL, internal standard (IS)), and the mixtures were vortex-mixed for 5 min, then centrifuged at 16,000× *g* for 5 min at 4 °C. The supernatants were analyzed using the Shimadzu 8040 Triple Quadrupole liquid chromatography-mass spectrometry (LC-MS/MS) system (Shimadzu, Kyoto, Japan). The separation was performed on a Kinetex XB-C18 column (2.1 mm × 150 mm, 2.6 μm, Phenomenex, Torrance, CA, USA) with a gradient elution of a mobile phase that consisted of acetonitrile and water with 0.1% formic acid (8% acetonitrile for 0–0.5 min, 60% acetonitrile for 5–6 min, 8% acetonitrile for 6.1–10 min) at a flow rate of 0.2 mL/min. The retention time was 3.18 min for acetaminophen, 5.08 min for hydroxydiclofenac, 4.56 min for hydroxymephenytoin, 4.02 min for dextrorphan, 4.67 min for 1′-hydroxymidazolam, and 5.35 min for propranolol (IS). The mass spectra were recorded by electrospray ionization with a positive mode. The quantification of metabolite formation was carried out using multiple reaction monitoring (MRM) at *m/z* 152 → 110 for acetaminophen, *m/z* 312 → 231 for hydroxydiclofenac, *m/z* 235 → 150 for hydroxymephenytoin, *m/z* 258 → 201 for dextrorphan, *m/z* 342 → 203 for 1′-hydroxymidazolam, and *m/z* 260.0 → 116.0 for propranolol.

### 2.3. Effect of P85 and Tween 80 on the P-gp-mediated Efflux in LLC-PK1-P-gp cells 

LLC-PK1-P-gp cells were seeded onto 24-well plates (BD-Corning, Corning, NY, USA) at a density of 1 × 10^5^ cells/well and grown for 24 h. After washing the cells twice with pre-warmed HBSS, the cellular accumulation of digoxin was initiated by the addition of 0.5 mL of HBSS containing 0.1 μM of [^3^H]digoxin and P85 (0.1% and 1%), tween 80 (0.1% and 1%), or cyclosporine A (25 μM, a representative P-gp inhibitor). Cyclosporine A stock solution in DMSO (10 mM) was 400-fold diluted in 0.5 mL of pre-warmed HBSS to make a final concentration of 25 μM of cyclosporine A. The reaction mixture was incubated for 30 min at 37 °C. After 30 min of incubation, the cells were washed three times with 1 mL of ice-cold HBSS and aspirated. The cells were then lyzed with 0.1 mL of 10% SDS solution, mixed with 0.5 mL Optiphase cocktail solution (Perkin Elmer Inc.; Boston, MA, USA), and stabilized for overnight incubation. The radioactivity of [^3^H]digoxin in the cell lysate was measured using a Microbeta 2 liquid scintillation counter (Perkin Elmer Inc.; Boston, MA, USA).

### 2.4. Optimization of Berberine Formulation

#### 2.4.1. Solubility Test

An excess amount of berberine chloride (10 mg) with various combinations of the berberine: P85: tween 80 mixtures (from 1:1:0.1 to 1:10:2) were incubated in 2 mL of distilled water for 4 h at 37 °C in a thermoshaker (300 rpm). Undissolved berberine powder was removed by centrifugation at 16,000× *g* for 20 min and filtered through a nylon membrane filter (0.45 μm) [26]. The berberine concentration in the filtrate was determined using a Shimadzu UV-1800 spectrophotometer (Shimadzu, Kyoto, Japan) at the maximum absorbance wave length of 346 nm.

#### 2.4.2. Size Distribution

Berberine-loaded P85 and tween 80 mixed micelles were prepared using the thin-film hydration method [27]. Briefly, berberine (50 mg) and a range of P85 (50–500 mg) and tween 80 (5–100 mg) were dissolved in methanol in a round bottom flask. The solvent was evaporated and the residual methanol remaining in the film was removed by freeze dryer (LABCONCO Co., Kansas city, MO, USA). The thin film was hydrated with 30 mL of distilled water at 37 °C for 2 h to obtain a micelle formulation, which was sonicated for 15 min, followed by centrifugation at 16,000× *g* for 20 min. The size distribution of the mixed micelle formulations of berberine: P85: tween 80 (from 1:1:0.1 to 1:10:1) was determined using Delsa Max Pro Light Scattering Analyzer (Beckman Coulter, Brea, CA, USA).

#### 2.4.3. Cytotoxicity

Caco-2 cells were seeded onto 96-well plates (BD-Corning, Corning, NY, USA) at a density of 1 × 10^5^ cells/well, and cultured for 24 h at 37 °C in a 5% CO_2_ incubator. The cells were exposed to 100 μL of culture medium containing various mixed micelle formulations of berberine: P85: tween 80 (from 1:5:0.1 to 1:10:0.5). As a control group, berberine was treated at concentrations of 0, 1, 5, 10, 50, 100, 500, and 1000 μM. After 24 h of treatment, 20 μL of CellTiter 96 aqueous one solution from a cell proliferation assay kit (Promega, Madison, WI, USA) was added to the culture medium and incubated for 1 h. The absorbance rate of the reaction mixture was measured at 492 nm using an Infiniti M200Pro microplate reader (Tecan, Zurich, Switzerland).

#### 2.4.4. Berberine Loaded Mixed Micelle Formulation

Berberine-loaded P85 and tween 80 mixed micelles with the ratio of 1:5: 0.5 were prepared using the thin-film hydration method [27]. Briefly, the methanol solutions of berberine (50 mg in 5 mL), P85 (250 mg in 10 mL), and tween 80 (25 mg in 5 mL) were added to a round bottom flask. The solvent was evaporated and the residual methanol remaining in the film was removed by a freeze dryer. The thin film was hydrated with 10 mL of distilled water at 37 °C for 2 h to obtain a micelle formulation, which was sonicated for 15 min, followed by centrifugation at 16,000× *g* for 20 min.

### 2.5. Caco-2 Permeability of Berberine Formulation 

Caco-2 cells were seeded onto filter inserts from 12-well transwell plates (BD-Corning, Corning, NY, USA) at a density of 5 × 10^5^ cells/well, and cultured for 24 h at 37 °C in a 5% CO_2_ incubator. The cells were grown for 21 days and the culture medium was changed every 2 days. The integrity of the cell monolayers was evaluated prior to the transport experiments by measuring the transepithelial electrical resistance (TEER). Values in the range of 300–650 Ω·cm^2^ were used in the transport experiment [28].

To characterize the absorption transport of berberine in Caco-2 cells, the apical to basal (A to B) and basal to apical (B to A) transport of berberine was measured. To measure the A to B transport of berberine, 0.5 mL of HBSS (pH 7.4) medium containing berberine (10 μM) in the presence or absence of cyclosporine A (25 μM) was added to the apical side, and 1.5 mL of HBSS medium without berberine was added to the basal side of the insert. The insert was then transferred to a well containing fresh HBSS medium every 15 min for 1 h. To measure the basal to apical (B to A) transport, 1.5 mL of HBSS medium containing berberine (10 μM) in the presence or absence of cyclosporine A (25 μM) was added to the basal side, and 0.5 mL of HBSS medium without berberine was added to the apical side. The transport medium on the apical side was replaced with 0.4 mL of fresh incubation medium every 15 min for 1 h. A 100 μL aliquot of each sample was stored at −80 °C for the analysis of berberine.

The concentration-dependent transport of berberine in Caco-2 cells was measured in a concentration range of 1–100 μM using a same procedure described above. A 100 μL aliquot of each sample was stored at −80 °C for the analysis of berberine.

To measure the A to B and B to A permeability of berberine and berberine formulation, 0.5 mL of HBSS supplemented with 10 mM of hydroxyethyl piperazine ethane sulfonicacid (pH 7.4) medium containing berberine (10, 25, 100 μM) or berberine formulation (equivalent to 10, 25, 100 μM of berberine) was added to the apical side, and 1.5 mL of HBSS medium without berberine was added to the basal side of the insert. The insert was then transferred to a well containing fresh HBSS medium every 15 min for 1 h. To measure the B to A transport, 1.5 mL of HBSS medium containing berberine (10, 25, 100 μM) or berberine formulation (equivalent to 10, 25, 100 μM of berberine) was added to the basal side, and 0.5 mL of HBSS medium without berberine was added to the apical side. The transport medium on the apical side was replaced with 0.4 mL of fresh incubation medium every 15 min for 1 h. A 100 μL aliquot of each sample was stored at −80 °C for the analysis of berberine. The TEER values were monitored before and after the transport experiments with berberine formulation using an epithelial TEER meter (World Precision Instruments, Sarasota, FL, USA).

A 100 μL aliquot of each sample was added to a 100 μL aliquot of ice-cold methanol containing 10 ng/mL of propranolol (IS). After vortexing and centrifuging the samples at 16,000× *g* for 10 min, the supernatant was injected directly into the LC-MS/MS system.

### 2.6. Intestinal Metabolism of Berberine Formulation

To examine the formation of berberine metabolites, berberine (10 μM) was incubated for up to 1 h in 0.1 M of KH_2_PO_4_ (pH 7.4) buffer that contained rat intestinal microsomes (0.25 mg/mL) and a NADPH generating system at 37 °C in a thermoshaker. All the reactions were terminated by the addition of 50 μL of ice-cold methanol containing 10 ng/mL of propranolol (IS), and were centrifuged at 16,000× *g* for 5 min at 4 °C. The supernatants were then analyzed via LC-MS/MS analysis.

To examine the effects of berberine formulation on the metabolism of berberine in intestinal microsomes, the formations of thalifendine and berberrubine, two major metabolites of berberine [8], were measured in the presence of berberine or berberine formulations (each well containing 10 μM of berberine). KH_2_PO_4_ buffer (0.1 M, pH 7.4, 90 μL) contained 0.25 mg/mL of protein from rat intestinal microsomes. The berberine or berberine formulations were preincubated for 5 min and the reaction mixtures were then incubated for another 15 min at 37 °C in a thermoshaker after a NADPH generating system (10 μL) had been added. All the reactions were terminated by the addition of 50 μL of ice-cold methanol containing 10 ng/mL of propranolol (IS), and were centrifuged at 16,000× *g* for 5 min at 4 °C. The supernatants were then analyzed via LC-MS/MS analysis.

### 2.7. Pharmacokinetics of Berberine 

#### 2.7.1. Animals

All the animal procedures were approved by the animal care and use committee of Kyungpook National University (Daegu, Korea; Approval No. KNU-2014-0097 and KNU-2019-0004). Male Sprague–Dawley rats weighing 250–270 g (8 weeks old) were purchased from Samtako Bio Korea (Osan, Kyunggido, Korea). When the rats arrived, they were housed with a 12 h light/dark cycle. Food and water were supplied ad libitum for 1 week prior to the animal experiment.

#### 2.7.2. Pharmacokinetic Study 

The rats were fasted for 12 h before the oral administration of the berberine or berberine formulation. The femoral vein, femoral arteries, and bile ducts of the rats were cannulated with polyethylene tubes (PE50 or PE10, Jungdo, Seoul, Korea) under anesthesia with isoflurane (30 mmol/kg) on the warming surgery table (YTK corp. Seoul, Korea), and the rats were remained on the warming table to maintain body temperature during the pharmacokinetic study. We suspended berberine chloride in a 0.5% carboxymethylcelluose suspension (50 mg/8 mL/kg of berberine), and the berberine formulation was hydrated in distilled water (50 mg/8 mL/kg of berberine). The prepared berberine suspension or berberine formulation was administered to rats via oral gavage. Blood samples (approximately 150 µL) were taken via the femoral artery at 0, 0.25, 0.5, 1, 1.5, 2, 4, 8, and 12 h after the oral administration, and pre-warmed saline (approximately 150 µL) was injected into the femoral vein to compensate the blood loss. The blood samples were centrifuged at 10,000× *g* for 1 min at 4 °C. Bile and urine samples were collected every 4 h for 12 h. After 12 h, the bile and urine samples were pooled and weighed. Aliquots (50 µL each) of plasma, urine, and bile samples were stored at −80 °C for the analysis of berberine and its metabolites.

Plasma, urine, and bile samples (50 µL) were added to 150 µL of methanol containing propranolol (10 ng/mL), vortex-mixed for 15 min, and centrifuged at 10,000× *g* for 5 min at 4 °C. An aliquot (10 µL) of the supernatant was injected into the LC–MS/MS system.

#### 2.7.3. LC-MS/MS Analysis of Berberine and Its Metabolites

Agilent 6430 Triple Quad LC/MS system (Agilent, Wilmington, DE, USA) coupled with an Agilent 1260 series HPLC system was used for the analysis of berberine and its metabolites using the modified method of Kwon et al. [1]. 

To identify the berberine metabolites in in vivo samples, 50 µL of bile samples were incubated with 100 µL of sodium acetate buffer (150 mM, pH 5.0) with or without 900 U of β-Glucuronidase and 30 U of sulfatase for 16 h at 300 rpm and 37 °C. The reaction was terminated by adding 150 µL of ice-cold methanol containing propranolol (10 ng/mL), vortex-mixed for 5 min, and centrifuged at 10,000× *g* for 5 min at 4 °C. An aliquot (10 µL) of the supernatant was injected into the LC–MS/MS system.

The separation was performed on a Synergy polar RP column (2.0 mm × 150 mm, 4 μm; Phenomenex, Torrance, CA, USA) using a gradient elution of the mobile phase that consisted of methanol and water with 0.1% formic acid at a flow rate of 0.2 mL/min. The composition of the mobile phase was as follows: 55% methanol for 0–2.5 min, 70% methanol for 3–5 min, 55% methanol for 5.3–12 min. The retention time was 3.01 min and 3.53 min for thalifendine-glucuronide and berberrubine-glucuronide, 6.30 min for thalifendine-sulfate, 7.79 min for berberine, 5.47 min for thalifendine, 6.38 min for berberrubine, and 2.53 min for propranolol. Mass spectra were recorded by electrospray ionization with a positive mode. Quantification was carried out using MRM at *m/z* 498.1 → 322.1 for thalifendine-glucuronide and berberrubine-glucuronide, *m/z* 402.0 → 322.1 for thalifendine-sulfate, *m/z* 336.1 → 320.0 for berberine, *m/z* 322.1 → 307.1 for thalifendine and berberrubine, and *m/z* 260.0 → 116.0 for propranolol. Calibration was applied on a standard curve in the range of 0.1–200 ng/mL of berberine, thalifendine, and berberrubine, respectively. The thalifendine-glucuronide, berberrubine-glucuronide, and thalifendine-sulfate were relatively quantitated using berberine’s standard curve, since the authentic standard compounds were not available. 

### 2.8. Data Analysis

The pharmacokinetic parameters were calculated by non-compartmental analysis using the WinNonlin Version 5.1 software (Pharsight, Certara, NJ, USA). Statistical analysis was performed using SPSS for Windows (version 24.0; IBM Corp., Armonk, NY, USA).

## 3. Results

### 3.1. Effect of P85 and Tween 80 on CYP Enzyme Activities and P-gp-Mediated Efflux 

We investigated the inhibitory effect of P85 and tween 80 on five major CYP enzyme activities in human liver microsomes and on P-gp efflux function using LLC-PK1-P-gp cells. For the system validation, representative inhibitors for the five CYPs and P-gp were used as a positive control (PC). The CYP1A2-mediated phenacetin O-deethylase, CYP2C9-catalyzed diclofenac 4-hydroxylase, CYP2C19-mediated mephenytoin 4-hydroxylase, CYP2D6-mediated dextromethorphan demethylase, and CYP3A-mediated midazolam 1-hydroxylase activities in ultra-pooled human liver microsomes were inhibited by the use of the PC. The use of P85 (0.1% and 1%) and tween 80 (0.1% and 1%) partially inhibited the activities of CYP1A2, CYP2C9, CYP2C19, CYP2D6, and CYP3A4 (Figure 2A–E). Among them, the CYP2D6 enzyme activity was the most significantly inhibited by the presence of P85 and tween 80. Similarly, the cellular accumulation of digoxin was increased by the presence of cyclosporine A (PC), P85, and tween 80 as a result of the P-gp-mediated digoxin efflux in the LLC-PK1-P-gp cells in a concentration-dependent manner (Figure 2F).

### 3.2. Optimization of Berberine-Loaded Mixed Micelle Formulation

#### 3.2.1. Berberine solubility

The berberine solubility was increased when the ratio of P85 to berberine was from 1:1 to 1:10 (*w/w*). The addition of tween 80 with a ratio of 0.1 and 0.5 (*w/w/w*) in the mixture of berberine: P85 from 1:1 to 1:10 greatly increased the solubility of berberine. However, the addition of tween 80 with a ratio over 1 gradually increased the solubility of berberine (Figure 3). Since the solubility was stable in the case of the berberine: P85 ratio of over 1:5 or 1:10 regardless of the tween 80 composition, we further investigated the optimal composition of tween 80 by investigating the size distribution and cytotoxicity of the triple mixture of berberine: P85: tween 80.

#### 3.2.2. Size Distribution of Berberine Formulation

The particle size of the berberine formulation dispersed in water may have been reduced by the addition of P85 with a ratio over 1:5. However, the presence of tween 80 greatly reduced the particle size of the berberine formulation even in the ratio over 0.1 of tween 80. The size distribution of the formulation was relatively stable, and there was less than 20 nm when the tween 80 content ratio was over 0.5 (Figure 4).

#### 3.2.3. Cytotoxicity in Caco-2 Cells

The cytotoxicity of P85 and tween 80 was investigated using the cell viability test in Caco-2 cells. Berberine itself did not induce cell toxicity at a dose of up to 100 μM, but the cell viability decreased with the increasing concentrations of berberine with the half maximal inhibitory concentration (IC_50_) of 933 μM. Berberine was also known for its anticancer effect in various cancer cells, which was mediated by the induction of apoptosis by the upregulation of p53 and the downregulation of Bcl-2, etc. [29,30]. These cytotoxic effects of berberine could contribute to the decreased cell viability of Caco-2 cells in the presence of high concentrations of berberine, and the IC_50_ value in Caco-2 cells were comparable with its IC_50_ value in human colon cancer HCT-8 cells (470 μM) [29]. 

We compared the IC_50_ values of berberine formulations with that of berberine itself (Figure 5). High concentrations of P85 and tween 80 decreased the cell viability, results in their IC_50_ values decreased to 319–467 μM when expressed as berberine concentration. However, the viability of the Caco-2 cells following the 24 h treatment of the berberine concentration of 100 μM with the berberine: P85: tween 80 of 1: 5: 0.5 or 1: 10: 0.1 was over 80%, suggesting that berberine formulations at this ratio could be used without inducing significant cytotoxicity at a berberine dose of up to 100 μM.

### 3.3. Inhibitory Effect of Berberine Formulation on P-gp-Mediated Berberine Efflux in Caco-2 Cells 

Based on the solubility, size distribution, and cell viability results, we selected a final berberine formulation containing a berberine:P85:tween 80 ratio of 1:5:0.5 (*w/w)* and investigated the effect of this berberine formulation on the P-gp-mediated berberine efflux and intestinal metabolism of berberine using Caco-2 cell monolayers and rat intestinal microsomes, respectively. 

Figure 6A shows that the B to A transport of berberine in rats was significantly greater than the A to B transport of berberine. The efflux ratio was calculated as 13.8; however, the presence of CsA dramatically decreased the B to A transport rate and the efflux ratio of berberine to 2.2. This suggests that berberine is a substrate for P-gp. The measurement of ATP consumption during the berberine transport in the presence or absence of CsA will confirm that berberine is a substrate for P-gp, considering the ATP dependency on the P-gp-mediated efflux [31].

Berberine absorption could, therefore, be disturbed by the presence of P-gp, and P-gp inhibition could impede the P-gp-mediated intestinal efflux of berberine. The B to A transport showed concentration dependency, and the nonlinear regression analysis using the Michaelis–Menten equation yielded such kinetic parameters as V_max_ and K_m_ values of 10.5 pmol/min and 6.5 μM, respectively. When compared to the clearance of berberine, the absorptive diffusion clearance was 0.03 μL/min, while the P-gp-mediated efflux clearance was calculated as 1.61 μL/min, which was 46-fold greater than the absorptive diffusion of berberine (Figure 6B). This suggests that the P-gp in Caco-2 cells plays a dominant role in the intestinal permeability of berberine. Next, we compared the Caco-2 permeability of berberine itself with that of the berberine formulation. When the same concentrations of berberine and berberine formulations were measured, the efflux ratio decreased. For example, the efflux ratio of 1 μM of berberine was 14.1, but this decreased to 6.33 when the same concentration was added to the berberine formulation. The efflux ratio of 100 μM of berberine was 7.5, but decreased to 1.1 when the same concentration was added to the berberine formulation (Figure 6C). This suggests that the P-gp modulation by highly concentrated berberine formulations was greater than that by low-concentration berberine formulations. Such P-gp modulations could be achieved by the use of P85 and tween 80, which was demonstrated in this study (Figure 2) and previous reports [17,32,33]. The TEER values were monitored before and after the transport study of the berberine formulation. As shown in Figure 6D, TEER values were not significantly altered by the treatment of a low dose of berberine formulation, but a high dose of berberine formulation significantly decreased the TEER value by 15.5–18.7%. These results suggested that the berberine formulation (at 100 μM as berberine) increased the cell permeability by partially disrupting tight junctions, which also could facilitate the penetration of berberine formulation in addition to its P-gp-inhibitory effect.

### 3.4. Inhibitory Effect of Berberine Formulation on Berberine Metabolism in the Rat Intestinal Microsomes

Thalifendine and berberrubine metabolites were formed from the rat intestinal microsomal incubation of berberine. In Figure 7A, the formation of thalifendine metabolite from berberine was much greater than that of berberrubine in rat intestinal microsomes, suggesting that thalifendine is a major metabolite in rat intestinal microsomes. A given concentration of berberine in berberine formulations significantly reduced the formation of thalifendine and berberrubine in comparison to the same concentration of berberine administered alone, but the fold decrease in thalifendine formation was much greater than that in berberrubine formation (Figure 7B). Taken together, the use of berberine formulations could inhibit the P-gp-mediated efflux and intestinal metabolism of berberine and, consequently, increase the intestinal absorption and plasma exposure of berberine.

### 3.5. Pharmacokinetics of Berberine and Its Metabolites in Berberine Formulaiton 

#### 3.5.1. Metabolite Identification in Triple Quadrupole Mass Spectrometry

Since the in vitro results indicated that berberine formulation could reduce the intestinal efflux and metabolism of berberine, we also investigated the pharmacokinetic alterations of berberine with the use of berberine formulations. 

In addition, we investigated the pharmacokinetics of berberine metabolites because the metabolic alteration in the intestinal absorption could alter the metabolism of berberine in vivo. Since we detected the formation of thalifendine and berberrubine metabolites from the intestinal microsomal incubation and Liu et al. [8] detected berberine, berberrubine, and thalifendine metabolites in rat plasma samples after the oral administration of berberine (50 mg/kg), we measured the berberine, thalifendine, and berberrubine in the rat plasma samples following the oral administration of berberine (50 mg/kg). As shown in Figure 8, the berberine (peak no. 4), thalifendine (peak no. 5), and berberrubine (peak no. 6) peaks were identified from the plasma samples, and the fragmentation patterns of these peaks were identical to those from the authentic standard compounds. We also identified two glucuronides (peak no. 1–2) and sulfate (peak no. 3) metabolites from the plasma samples. 

To elucidate the structure of the glucuronide and sulfate metabolites, bile samples collected for 12 h after the oral administration of berberine samples were treated with glucuronidase and sulfatase and compared with control bile samples (Figure 9). After the glucuronidase and sulfatase treatment, two glucuronide peaks and one sulfate metabolite had disappeared, while the thalifendine peak increased by 10.6-fold. There was no significant change in the berberine peaks (Figure 9A,B). Berberrubine was not detected in the bile samples from both the control and glucuronidase treatment groups (Figure 9B,C). When the stability of berberine, thalifendine, and berberrubine following 8 h of incubation of glucuronidase and sulfatase was measured, the berberine and thalifendine were stable but more than 94% of the berberrubine was degraded by the glucuronidase and sulfatase treatment (Figure 9D), suggesting that even berberrubine formed from glucuronide metabolites could be degraded by the glucuronidase and sulfatase treatment. Wang et al. [34] and Xu et al. [9] also identified two demethylberberine-glucuronide and one demethylsulfate in rat urine, plasma, bile, and feces samples, which were identified as thalifendine-glucuronide, berberrubine-glucuronide, and thalifendine/berberrubine-sulfate based on the MS^2^ spectra of the berberine metabolites by UPLC-Q-TOF-MS and a metabolite identification program. The MS^2^ spectra and retention time of the phase II metabolites were identical to the MS/MS fragmentation pattern in this study (Figure 9C). Taken together, the three phase II metabolites in this study could be identified as thalifendine-glucuronide, berberrubine-glucuronide, and 1 thalifendine/berberrubine-sulfate (Figure 10), which has also been previously reported [9,35,36,37]. 

#### 3.5.2. Pharmacokinetics of Berberine and Its Metabolites Following the Oral Administration of Berberine Formulaiton

Figure 11 shows the berberine plasma concentrations vs. time profiles of berberine and its metabolites in rat plasma samples following the oral administration of berberine alone and berberine formulations (50 mg/kg each as berberine). Among the berberine metabolites, thalifendine-glucuronide (Figure 11D) were the highest metabolite found in plasma samples, and the AUC values of the thalifendine conjugate metabolites were greater than those of thalifendine and berberrubine, suggesting the faster phase II conjugation of thalifendine from thalifendine in both the berberine and berberine formulation groups (Figure 11 and Table 1). In addition, considering the plasma concentration of thalifendine and thalifendine conjugates, thalifendine seemed to be a major metabolite compared with the AUC values of berberrubine or berberrubine-glucuronide, which is consistent with previous reports [9]. 

The plasma concentrations of berberine and its metabolites were higher in the berberine formulation group than in the berberine group. The fold increase in the berberine AUC in the berberine formulation group was 15.6-fold compared with that of the berberine group. Similarly, the AUC values of thalifendine and thalifendine-sulfate were significantly greater than in the berberine group by 9.3- and 5.8-fold, respectively (Figure 11A,B,F). The fold changes in the AUC values of berberrubine and thalifendine from rats orally administered berberine formulations were not significantly altered compared with rats administered berberine (Figure 11C,D,E). The total amount of excreted berberine and its metabolites in urine and bile for 12 h was significantly greater in rats administered the berberine formulations than in rats administered the berberine itself (Table 1). This result suggests that the berberine absorption increased when the berberine formulation was used. In addition, the metabolic ratios of thalifendine conjugates or berberrubine following the administration of the berberine formulations were significantly decreased compared with those following administering berberine itself (Table 1), which could be attributed to the inhibitory effect of the P85 and tween 80 excipients used in the berberine formulations on the berberine metabolism.

## 4. Discussion

Berberine has long been used against diarrhea and microbial activities in the gut, and has had a good clinical efficacy [11]. Berberine also has therapeutic potential for its antidiabetic and antihyperlipidemic effects [12]. However, berberine treatment has not been widely used for the therapeutic goal of lowering glycemic and cholesterol levels, as plasma concentrations of berberine are very low and rapidly eliminated [12,38]. When orally administered, only 0.5% of a berberine dose could enter the portal vein. About 56% of the berberine oral dose was not absorbed because of the P-gp-mediated efflux and self-aggregation of berberine [12]. About 43.5% of a berberine oral dose was metabolized in the enterocytes [12]. Therefore, increased berberine absorption could be a useful strategy for the development of berberine therapeutics as antidiabetic or antilipidemic agents, since berberine was highly distributed to the liver (i.e., the AUC_liver_ of berberine 728.6 ng∙h/mL vs. the AUC_plasma_ of berberine 86.4 ng∙h/mL) [38].

In this study, we developed berberine-loaded mixed micelle formulation ratios of berberine: P85: tween 80 at 1: 5: 0.5 (*w/w/w*) using a thin-layer film hydration method. This berberine formulation increased the berberine solubility by 800% and had a mean particle size of 12 nm, without significant cytotoxicity for up to 100 μM (Figure 5). P85 (molecular weight (mw) of 4600) and tween 80 (mw of 1310) are large mw surfactants, and their mixed micelles incorporating berberine are also large mw formulations in the intestinal fluids. Once the berberine-loaded mixed micelles contact the intestinal membrane, they could decrease the cell integrity and increase the paracellular pathway because of the surfactant effect of P85 and tween 80. This is evidenced by the decrease in the TEER value (Figure 6D) and also in our previous report [39].

Lo et al. [40] reported that the decrease in TEER values by non-ionic surfactants such as tween 80 was reversible. Park et al. [41] reported that surfactants with permeability-enhancing effects and a medium fatty acid chain length may penetrate the lipid bilayer easily because of their proper lipid solubility. Qu et al. [42] reported the enhanced permeability of paclitaxel in Caco-2 cells using paclitaxel-loaded mixed polymeric micelles combining poly (2-ethyl-2-oxazoline)-vitamin E succinate with TPGS, in which the energy-dependent transmembrane transport of a mixed micelle formulation was involved via both clathrin- and caveolae-mediated micropinocytosis mechanisms. Taken together, a mixed micelle formulation consisting of a large mw surfactant could penetrate the intestinal lumen through its decreased cell integrity, which was evidenced by the decrease in the TEER values and/or micropinocytosis transmembrane mechanisms [27,39,42]. Our berberine-loaded mixed micelle formulation could act as a P-gp modulator and CYP inhibitor in the enterocytes through the decreased cell membrane integrity by increasing the modulation of tight junctions in parts. A more detailed mechanism study regarding the transmembrane process of the berberine formulation needs to be performed.

The berberine mixed micelle formulation increased the absorption permeability of berberine by 364%, and the efflux ratio (i.e., the ratio of efflux permeability to absorptive permeability) decreased from 7.54 to 1.05 (Figure 6C), suggesting that the P-gp-mediated efflux and absorption enhancement of berberine was inhibited. In addition, thalifendine, a major metabolite of berberine in the intestinal microsomes, decreased by 76% after the addition of the berberine formulation compared with the thalifendine formation from berberine alone (Figure 7B). CYP1A2, CYP2D6, and CYP3A4 have been reported to be involved in the berberine metabolism [9]. P85 and tween 80 have been shown to have inhibitory potential for CYP2D6 and CYP3A4 (Figure 2). In addition, CYP2D6 and CYP3A are expressed in the rat intestines, and contribute to the intestinal first-pass metabolism of many drugs [43,44,45]. Collectively, by using berberine: P85: tween 80 mixed micelle formulations, the intestinal absorption of berberine could be increased due to the inhibition of the P-gp-mediated efflux and intestinal metabolism, which accounts for the major intestinal first-pass effects on berberine [8,12]. This in vitro inhibitory effect of berberine formulation on intestinal efflux could be caused by the metabolism and increased solubility, which could contribute to the pharmacokinetics and metabolism of berberine. To test this, we compared the pharmacokinetics of berberine and its major metabolites following the berberine formulation (50 mg/kg as berberine content) with those of berberine itself (50 mg/kg) in rats. At first, thalifendine and thalifendine-glucuronide were much higher than berberrubine and the secondary metabolites from berberrubine in rat plasma. This result suggests that thalifendine and thalifendine-glucuronide are major metabolites of berberine (Figure 11), which was consistent with previous reports [10,11]. Second, the AUC values of berberine and its metabolites (thalifendine and thalifendine conjugates) were greatly increased by the use of berberine formulations. When X_u,12h_ and X_b,12h_ are compared (Table 1), the sum of berberine and its metabolites in the urine and bile of rats following the administration of the berberine formulation was significantly higher than when only berberine was administered, which suggests that the berberine formulation increases berberine absorption. The increased plasma concentrations of thalifendine and thalifendine conjugates could be explained as the result of the increased plasma concentrations of berberine. To differentiate the effect of excipients such as P85 and tween 80 on berberine metabolism, we calculated the MR of berberine metabolites to berberine (Table 1). We found that the metabolic ratios of berberrubine and thalifendine conjugates were greatly decreased by the use of the berberine formulation. In addition, the MR of thalifendine-glucuronide was decreased 2.1-fold, but the MR of berberrubine-glucuronide was decreased 8.3-fold. The results indicated that berberine metabolism was inhibited by the use of P85 and tween 80, but that the inhibitory effect of these excipients was different depending on the thalifendine or berberrubine pathway.

Collectively, our berberine-loaded mixed micelle formulation could increase berberine plasma concentrations by increasing berberine intestinal absorption through the use of pharmaceutical excipients that inhibit P-gp function and berberine metabolism. This could contribute to the beneficial effect of berberine therapy in regard to antidiabetic effects and antihyperlipidemic actions by increasing the plasma exposure of berberine.

## Figures and Tables

**Figure 1 pharmaceutics-12-00882-f001:**
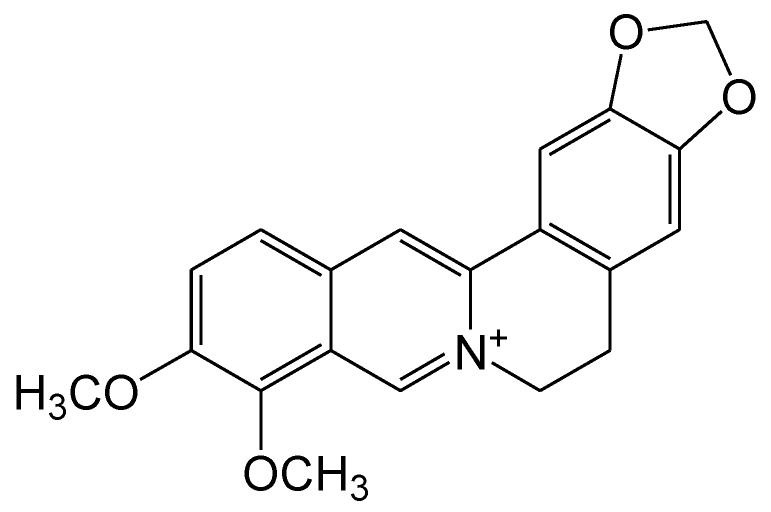
Structure of berberine.

**Figure 2 pharmaceutics-12-00882-f002:**
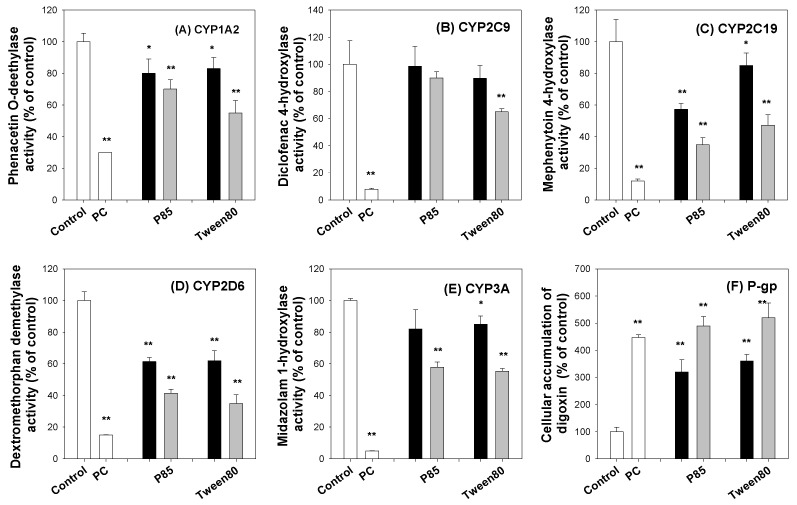
Inhibitory effect of positive control (PC), pluronic P85 (P85, black bar: 0.1% and gray bar: 1%) and tween 80 (black bar: 0.1% and gray bar: 1%) on (**A**) CYP1A2-mediated phenacetin *O*-deethylase, (**B**) CYP2C9-catalyzed diclofenac 4-hydroxylase, (**C**) CYP2C19-mediated mephenytoin 4-hydroxylase, (**D**) CYP2D6-mediated dextromethorphan demethylase, and (**E**) CYP3A-mediated midazolam 1-hydroxylase activities in ultra-pooled human liver microsomes in the presence of NADPH regenerating systems at 37 °C. (**F**) Inhibitory effects of P85 and tween 80 on the P-gp-mediated efflux of digoxin in LLC-PK1 cells that overexpress P-gp. Representative inhibitors for CYP isozymes and P-gp were used as PCs: 0.1 μM of α-Naphthoflavone (CYP1A2), 5 μM of sulfaphenazole (CYP2C9), 5 μM of S-benzylnirvanol (CYP2C19), 5 μM of quinidine (CYP2D6), 5 μM of ketoconazole (CYP3A), 25 μM of cyclosporine A (P-gp). The substrate concentrations for the CYP isozymes and P-gp were as follows: 50 μM of phenacetin (CYP1A2), 10 μM of diclofenac (CYP2C9), 100 μM of mephenytoin (CYP2C19), 5 μM of dextromethorphan (CYP2D6), 5 μM of midazolam (CYP3A), and 1 μM of [^3^H]digoxin (P-gp). The data are expressed as means ± standard deviations (*n* = 3). *: *p* < 0.05 and **: *p* < 0.01; statistically significant compared with control group by Student’s *t*-test.

**Figure 3 pharmaceutics-12-00882-f003:**
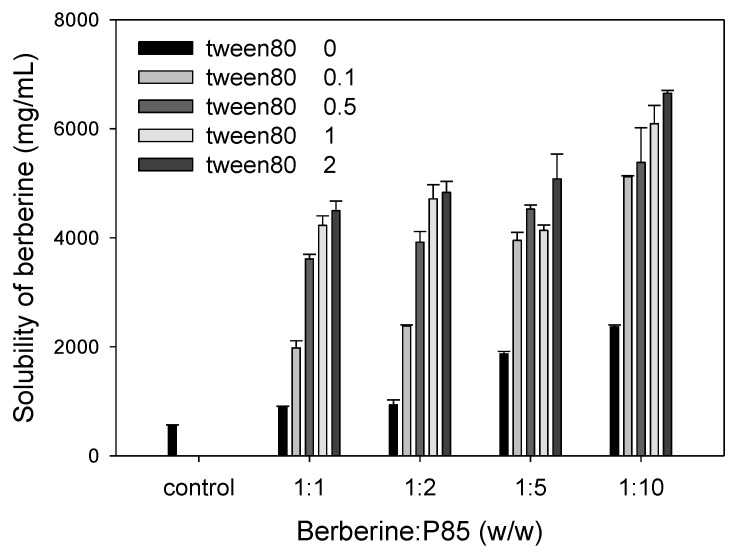
Solubility of berberine was determined by increasing the ratio of P85 and tween 80 (*w/w/w*) compared to berberine using a UV spectrophotometer. Each bar represents the mean ± standard deviation (*n* = 3).

**Figure 4 pharmaceutics-12-00882-f004:**
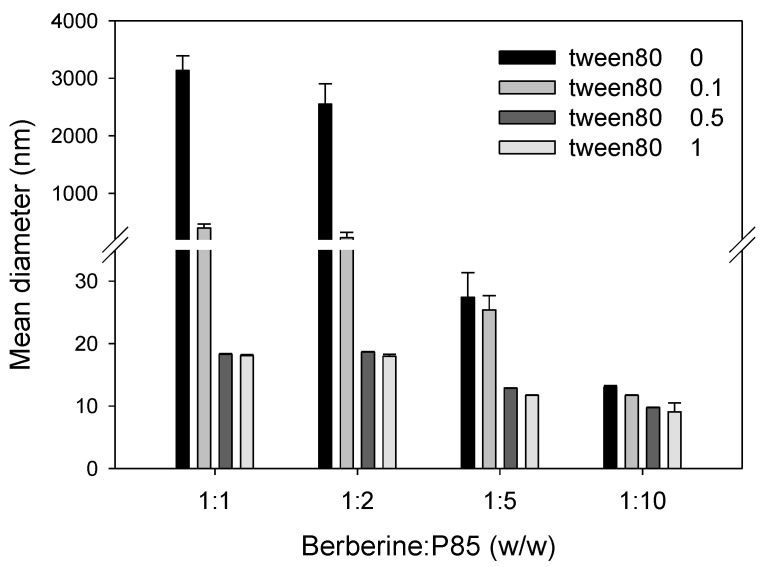
Size distribution of the berberine formulation with an increasing ratio of P85 and tween 80 (*w/w/w*) compared to berberine. Bar represents the mean ± standard deviation (*n* = 3).

**Figure 5 pharmaceutics-12-00882-f005:**
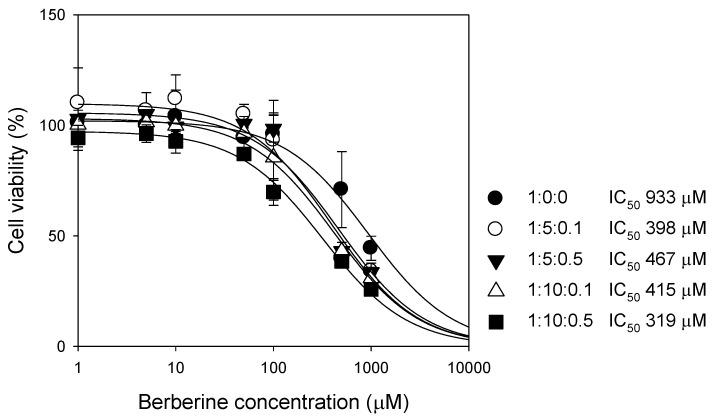
The effect of berberine or berberine formulations on cell viability in Caco-2 cells. Cell viability was monitored after the 24 h incubation of berberine only (●) or various compositions of berberine: P85: tween 80 (*w/w/w*) (○, ▼, △, □), which were expressed as molecular concentrations of berberine. Each data point represents the mean ± standard deviation (*n* = 3).

**Figure 6 pharmaceutics-12-00882-f006:**
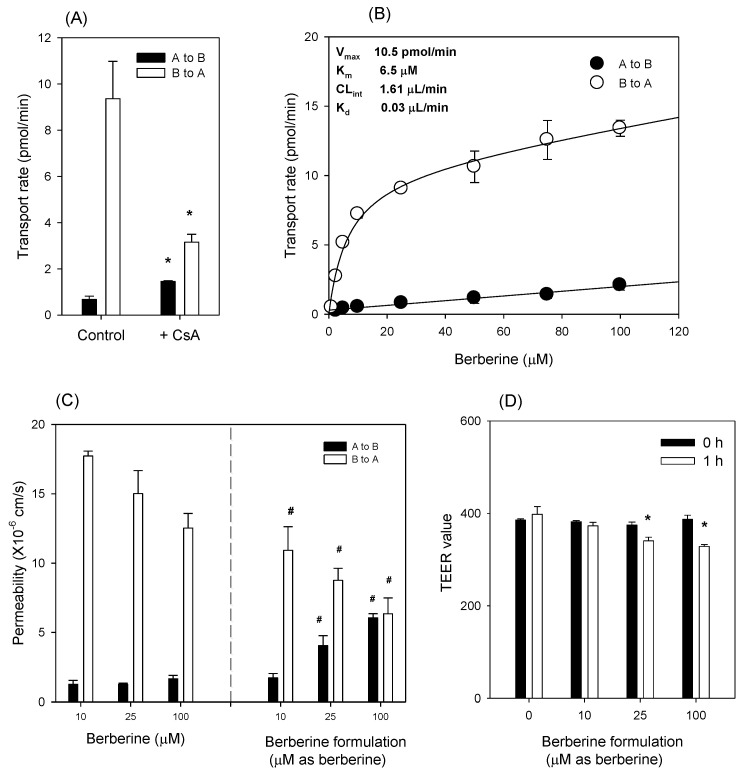
(**A**) Apical to basal (A to B; closed symbol) and basal to apical (B to A; open symbol) transport of berberine (10 μM) in Caco-2 cells in the presence or absence of cyclosporine A (CsA, 25 μM). (**B**) Concentration dependence in the A to B and B to A transport rate of berberine in Caco-2 cells. (**C**) A to B and B to A permeability (P_app_) of berberine or berberine formulation in Caco-2 cells. (**D**) Transepithelial electrical resistance (TEER) values (Ω·cm^2^) were monitored before (0 h) and after (1 h) the transport study of berberine formulation. Each data point represents the mean ± standard deviation (*n* = 3). *: *p* < 0.05, statistically significant compared with the control group by Student’s *t*-test. #: *p* < 0.05, statistically significant compared with berberine itself by Student’s *t*-test.

**Figure 7 pharmaceutics-12-00882-f007:**
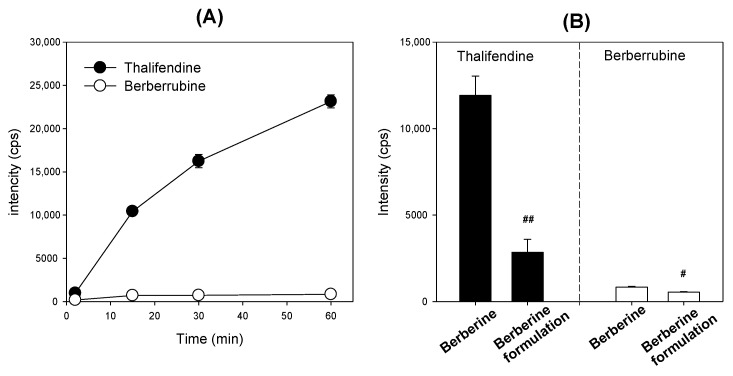
(**A**) Formation of thalifendine (●) and berberrubine (○) metabolites from berberine in rat intestinal microsomes. (**B**) Formation of thalifendine (●) and berberrubine (○) metabolites from berberine or berberine formulation in rat intestinal microsomes. Each data point represents the mean ± standard deviation (*n* = 3). #: *p* < 0.05 and ##: *p* < 0.001, statistically significant compared with berberine itself group by Student’s *t*-test.

**Figure 8 pharmaceutics-12-00882-f008:**
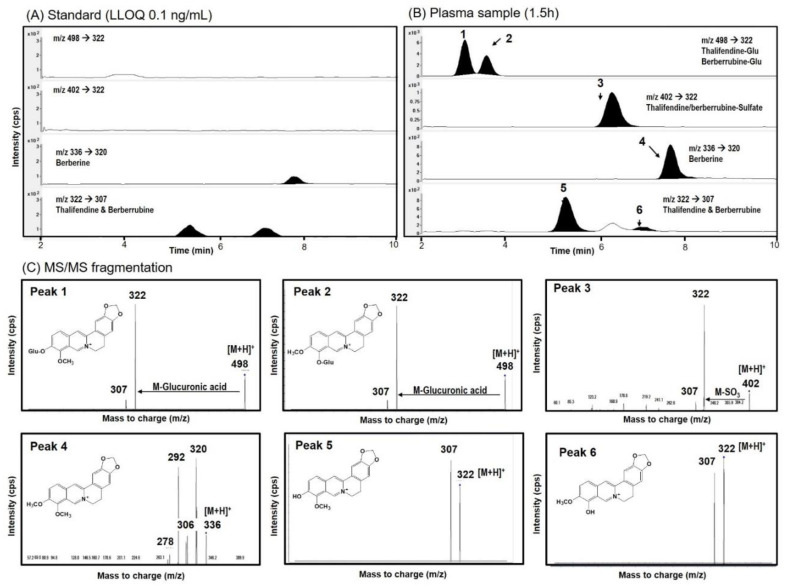
(**A**) Representative multiple reaction monitoring (MRM) chromatograms of berberine, thalifendine, and berberrubine at a concentration of 0.1 ng/mL (lower limit of quantification, LLOQ). (**B**) Representative MRM chromatograms and (**C**) MS/MS fragmentation pattern of berberine (peak no. 4) and elucidated berberine metabolites (peaks 1–3, 5, 6) in the plasma samples at 1.5 h after the oral administration of berberine.

**Figure 9 pharmaceutics-12-00882-f009:**
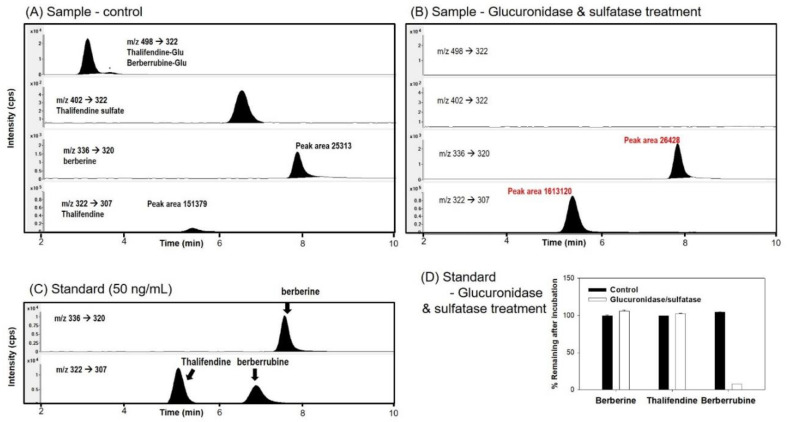
MRM chromatograms of berberine and berberine metabolites in the bile samples collected for 12 h after the oral administration of berberine with (**B**) or without glucuronidase and sulfatase treatment (**A**). (**C**) MRM chromatograms of berberine, thalifendine, and berberrubine standard spiked in blank bile. (**D**) Stability of berberine, thalifendine, and berberrubine after the glucuronidase and sulfatase treatment for 16 h.

**Figure 10 pharmaceutics-12-00882-f010:**
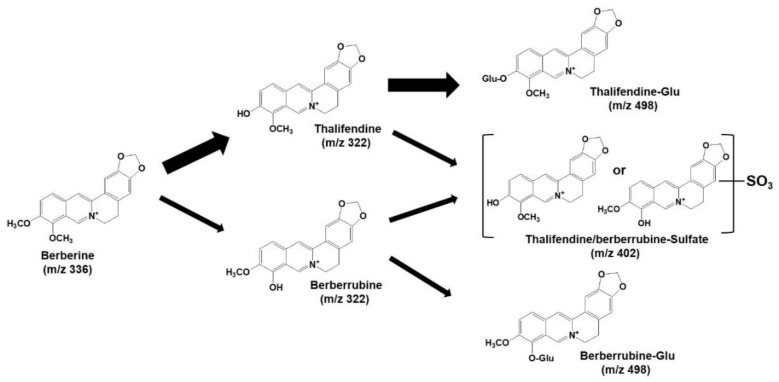
Identified major metabolites of berberine in rat plasma, urine, and feces samples following the oral administration of berberine or berberine formulation. Glu: glucuronide.

**Figure 11 pharmaceutics-12-00882-f011:**
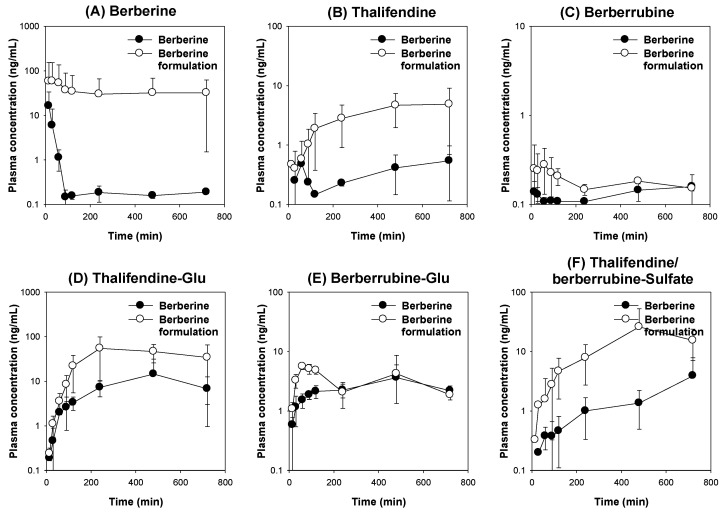
Plasma concentration–time profile of (**A**) berberine, (**B**) thalifendine, (**C**) berberrubine, (**D**) thalifendine-glucuronide, (**E**) berberrubine-glucuronide, and (**F**) thalifendine/berberrubine-sulfate following the oral administration of berberine or berberine formulations at a dose of 50 mg/kg in rats. Data represent means ± standard deviations (*n* = 3). Glu; glucuronide.

**Table 1 pharmaceutics-12-00882-t001:** Pharmacokinetic parameters of berberine and its metabolites following the oral administration of berberine or berberine formulation at a dose of 50 mg/kg in rats.

Administration		Berberine (PO, 50 mg/kg)					
Parameters	Unit	Berberine	Thalifendine	Berberrubine	Thalifendine-Glucuronide	Berberrubine-Glucuronide	Thalifendine/Berberrubine-Sulfate
T_max_	min	30.0 ± 26.0	420.0 ± 334.1	250.0 ± 407.0	420.0 ± 334.1	430.0 ± 318.0	640.0 ± 138.6
C_max_	ng/mL	67.0 ± 66.9	0.64 ± 0.19	0.16 ± 0.03	14.0 ± 11.3	3.31 ± 1.71	3.26 ± 3.03
AUC	ng·min/mL	2067 ± 1731	182.4 ± 83.6	80.4 ± 24.4	4260 ± 1680	1549 ± 750	1025 ± 1033
Metabolic ratio	%		0.09	0.04	2.06	0.75	0.50
X_b,12h_	ng	323.7 ± 155.8	281.5 ± 250.2	0	311.3 ± 226.5	44.19 ± 51.71	41.93 ± 25.77
X_u,12h_	ng	47.3 ± 30.1	199.3 ± 76.1	22.80 ± 15.84	747.2 ± 529.6	613.8 ± 775.0	49.86 ± 36.48
**Administration**		**Berberine Formulation (PO, 50 mg/kg)**					
**Parameters**	**Unit**	**Berberine**	**Thalifendine**	**Berberrubine**	**Thalifendine-Glucuronide**	**Berberrubine-Glucuronide**	**Thalifendine/Berberrubine-Sulfate**
T_max_	min	35.0 ± 22.9	640.0 ± 138.6	55.00 ± 37.75	400.0 ± 277.1	210.0 ± 234.3	640.0 ± 138.6
C_max_	ng/mL	140.3 ± 29.4	5.45 ± 3.12	0.31 ± 0.18	56.3 ± 41.8	6.08 ± 1.22	25.2 ± 18.0
AUC	ng·min/mL	32,214 ± 18,966 #	1688 ± 860 #	104.1 ± 45.3	20,371 ± 12,753 ##	1813 ± 184	5988 ± 2624 #
Metabolic ratio	%		0.08	0.00	0.96	0.09	0.28
X_b,12h_	ng	992 ± 563	1871 ± 1092 #	0	2412.4 ± 1351.2 #	84.38 ± 28.56	411.5 ± 395.0
X_u,12h_	ng	21,273 ± 12,517 #	7209 ± 2632 ##	67.78 ± 11.89 #	16,471 ± 9413 #	1721 ± 260	1042 ± 225 ##

Abbreviations: PO, per oral administration; C_max_, maximum plasma concentration; T_max_: time to reach C_max_; AUC, area under the curve; metabolic ratio, ratio of metabolite AUC to berberine AUC; X_b,12h_, amount excreted into bile for 12 h; X_u,12h_, amount excreted into urine for 12 h. #: *p* < 0.05 and ##: *p* < 0.01, statistically significant compared with the berberine group by Student’s *t*-test. Data represent means ± standard deviations (*n* = 3).

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
