# Peer review of "Enhanced Intestinal Absorption and Pharmacokinetic Modulation of Berberine and Its Metabolites through the Inhibition of P-Glycoprotein and Intestinal Metabolism in Rats Using a Berberine Mixed Micelle Formulation"

_pharmaceutics, 2020, doi:10.3390/pharmaceutics12090882_

Round 1
Reviewer 1 Report
To enhance the oral bioavailability, authors prepared berberine mixed micelle formulations containing P-glycoprotein inhibitor, and the oral bioavailability of berberine was greatly enhanced in rat study. Although experimental results are interesting, some corrections are necessary, I think. Please check the following comments.
<Major criticism> As the mechanism, authors described that the enhanced oral bioavailability of berberine through the use of mixed micelle formulation was due to the inhibition of intestinal P-gp-mediated efflux transport and intestinal metabolism (Abstract and Conclusion,Line 549-). However, both surfactants have large molecular weights. Also they form mixed micelles with very large molecular weights in the intestinal fluids. Such compounds will not permeate plasma membrane, nor reach to the microsomal fraction in enterocytes without causing membrane damage.
<Minor criticism>
- More detailed information is necessary about experiments. For example, was free base berberine used in the present study? In Sigma-Aldrich, berberine chloride, berberine chloride hydrate, berberine hemisulfate salt are available. How did you dissolve berberine, a poorly water soluble, at a concentration of 1 mM or 50 mg/kg for animal study? Did you use free base quinidine? Cyclosporine A is almost insoluble in water. How did you dissolve cyclosporine A for transport study of berberine? Please check whole experimental section carefully.
- In solubility study, to separate undissolved fraction of berverine, the suspensition was centrifuged at 16,000g for 20 min. Is this method OK to separate precipitation? Please cite an appropriate reference about this in the text.
- Regarding the animal study: More detailed information such as the volume of blood sampled each time, control method of body temperature during absorption study, hydration method, etc. is required, because blood and/or bile juice sampling may reduce the blood flow rate in the body and fluctuate hepatic and renal clearance.
Author Response
Thank you for the reviewer's valuable comments.
According to the reviewers’ comment, we added the more detailed experimental description regarding the pharmacokinetic study and inhibition study. We also added the results and discussion on the cell membrane integrity by measuring TEER value before and after the transport of berberine formulation in Caco-2 cells during the revision. Other comments were reflected accordingly during the revision and corrections were highlighted in blue.
Our point-by-point response to the reviewer's comments was attached.

Reviewer 2 Report
pharmaceutics-926511 describes the design and optimization of a berberine mixed micelle formulation in order to increase absorption and reduce metabolism, through the inhibition of Pgp-mediated efflux and intestinal metabolism.
The approach and results obtained are interesting; the study is rational and well written. However I suggest some corrections to make the manuscript more usable.
- On page 2 line 90: “Tween 80 is also well known for the inhibition of P-gp inhibition” maybe there is a repetition
- On page 2 lines 91-92: I'm not sure if 24 is the correct reference. The article describes berberine and its metabolites and not tween 80 and digoxin.
- On page 9 section 3.2.3: The authors say that berberine does not induce cell toxicity but decreases the cell viability. Are there any possible explanations for this behavior?
- On page 10 lines 365-370: Based on the value of the efflux ratio, it can be asserted that berberine is a substrate for P-gp. Did the authors measure ATP consumption? Based on the consumption of ATP, it can thus be confirmed that berberine is a substrate of P-gp and to exclude that it is a non-transported substrate. See L. Kangas, M. GrÓ§nroos, A.L.Nieminem, (1984), Med. Biol. 62, 338-343.
- On page 1 line 385: (Figure 1) is correct?
Author Response
Thank you for the reviewer's valuable and positive comments.
We revised our manuscript carefully according to the reviewer’s comments and corrections were highlighted in blue.
Our point-by-point response to the reviewer's comments was attached.

Round 2
Reviewer 1 Report
Appropriate explanations were made to the points that I previously pointed out as lack of explanation in the revised manuscript, I think.